# Risk factors and visual outcome of Non-Arteritic Ischemic Optic Neuropathy (NAION): Experience of a tertiary center in Kuwait

**Raed Behbehani** *, **Abdullah Ali, Ashref Al-Moosa**

Al-Bahar Ophthalmology Center, Ibn Sina Hospital, Kuwait, Kuwait

* rsbehbehani@gmail.com

## Abstract

### Background

Non-arteritic ischemic optic neuropathy (NAION) is the most common acute optic neuropathy over the age of 50 years. NAION is commonly associated with systemic vascular risk factors (diabetes, hypertension, hyperlipidemia) and small cup-to-disc-ratio. We have assessed the prevalence risk factors of NAION and the visual outcome in patients referred to a tertiary ophthalmology center in Kuwait.

### Materials and methods

A retrospective review of new cases of NAION presenting within 2 weeks of onset were included and baseline clinical and demographics characteristic were determined. The prevalence of risk factors and the visual outcome (change in logMAR visual acuity, mean deviation of visual field) was compared between young NAION patients (below 50 years of age) and older NAION patients (over 50 years of age). The odds ratio of a final favorable visual outcome (visual acuity 20/40 or better) by age category was determined.

### Results

Seventy-eight eyes of 78 patients with recent onset NAION were included in the study. The most prevalent risk factors for NAION in our subjects were diabetes (64.1%), small cup-to-disc ratio (61.5%), hyperlipidemia (51.3%) and hypertension (38.5%). Young NAION patients had better final logMAR visual acuity (0.55 +- 0.57) then older NAION patients (0.9 +- 0.73), (p = 0.03). Furthermore, young NAION patients were 2.8 times more likely to have a final visual acuity of 20/40 or better than older NAION patients, odds ratio (OR), 2.87; 95% confidence interval (CI), 1.12–7.40, Chi-square p-value = 0.03.

### Conclusion

There is a high prevalence of systemic vascular risk factors and small cup-to-disc ratio in NAION patients referred to our center across different age groups (below and above 50 years). Patients below the age of 50 years with NAION are more likely to have a final visual acuity of 20/40 or better than NAION patients above the age of 50 years.

**Data Availability Statement:** All relevant data are within the manuscript.

**Funding:** I have received no funding for this work and I have no financial interests to disclose.

**Competing interests:** The authors have declared that no competing interests exist.

## Introduction

Non-arteritic ischemic optic neuropathy (NAION) is the second most common optic neuropathy following glaucoma in patients over the age of 50 years and is the most common cause of optic nerve-related acute vision loss [1]. It is associated with systemic risk factors such as diabetes, hypertension, hyperlipemia and anatomical risk factors mainly crowded anomalous optic disc (disc at risk) or small cup-to-disc ratio [2,3]. The exact mechanism of NAION is not fully understood but it is thought to result from hypo-perfusion of the optic nerve head resulting in optic nerve head ischemia and swelling. Since predisposed individual frequently have a small cup-to-disc ratio, this can lead to a secondary compartment syndrome and further ischemia [4]. There are few studies that examined the natural history and the visual outcome of NAION [1,5,6]. There is increasing awareness that NAION can also occur in younger patients and is occasionally misdiagnosed as optic neuritis or papillitis, which can lead to inappropriate work-up and management [7,8]. In this study we have investigated the clinical characteristics and the visual outcome of patients with NAION, who were referred to a tertiary ophthalmology center. Moreover, we have assessed the visual outcome of NAION patients below 50 years of age in comparison to NAION patients above the age of 50 years.

## Methods

We have retrospectively reviewed the records of patients presenting with a diagnosis of NAION between the period of 2006–2019. All patients were assessed by two neuro-ophthalmologists (RB, AM) and only new NAION cases which presented within 2 weeks of symptom onset were included. The diagnosis was based on the typical clinical presentation of sudden painless visual loss with a relative afferent pupillary defect and a swollen optic disc with or without disc hemorrhages. Additional supportive findings included the presence of a crowded disc or a small cup-to-disc ratio in the same eye or the contralateral eye, and the presence of the classical systemic risk factors (diabetes, hypertension, and hyperlipidemia). Patients with previous NAION in the contralateral eye, perioperative ischemic optic neuropathy (ocular or systemic surgery), shock-induced ischemic optic neuropathy and patients with severe non-proliferative diabetic retinopathy, proliferative diabetic retinopathy or diabetic macular edema, were excluded. Medical records were reviewed to obtain clinical date including systemic risk factors identified either by direct history from the patient or laboratory testing. Snellen visual acuity was obtained at presentation and at the final follow up and was converted into logarithm of the minimum angle of resolution (logMAR) visual acuity to aid statistical analysis. A change in visual acuity of at least 0.3 logMAR unit change was set as a criterion for either improvement or worsening and a change in visual field mean deviation (MD), whether improvement or worsening, was defined as a change of at least 3 decibels.

Vertical cup-to-disc ratio was obtained using either time-domain optical coherence tomography for patients seen between 2006–2008 (TD-OCT, Stratus OCT, Carl Zeiss Meditec, USA), and with spectral domain optical coherence tomography (Topcon SD-OCT 3000) for patients seen from 2008 onwards. The visual field MD was obtained from Humphrey automated perimetry (24–2 FAST SITA strategy) at both baseline and the last follow up. The study was approved by the research ethics committee of Al-Bahar Ophthalmology Center.

## Statistical analysis

Descriptive statistics were applied for demographic clinical variable and Chi-square test was used to study the association between categorical variables. To compare the visual outcome of NAION in patients below and above 50 years of age, Mann–Whitney U test was used for the

change in logMAR visual acuity and MD of visual fields from baseline to the last recorded visit.

Multivariable binary logistic regression was used to determine if there was any independent effect of the NAION risk factors on the visual outcome. Statistical analysis was done using SPSS (IBM Corp. IBM SPSS, Version 23.0).

## Results

### Clinical characteristics of NAION patients

Seventy-eight eyes of 78 patients with newly diagnosed NAION were included in the study. The baseline characteristics of the patients are shown in Table 1.

Our subjects were of various ethnic backgrounds but were predominantly Middle-Eastern Arabs (75%) followed by Indian (19%) and Asian (6%) backgrounds. The most prevalent risk factors for NAION were diabetes, cup-to-disc ratio equal to or below 0.3, hyperlipidemia and hypertension. A significant proportion of our study subjects (41%) were below the age of 50, and a cup-to-disc ratio ($\leq$ 0.3) was found in 68.75% of young NAION patients and 56,6% of older NAION patients. Sixty-four percent of our subjects had at least 3-month follow up while the rest (31%) had at least 2-month follow up and only four patients (5%) had 1-month follow up.

NAION subjects below 50 years of age were composed mainly of males (84.4%) while in NAION over 50 years of age, gender distribution was more equal with 56.6% males and 43.5%

**Table 1. Baseline clinical characteristics of study subjects.**

| Patients (n = 78) | | Mean ± standard deviation |
|---|---|---|
| **Age in years, mean± SD** | | 50.60 (± 9.6) |
| **Age category** | **Below 50 years (n,%)** | 32 (41.0%) |
| | **Above 50 years (n,%)** | 46 (59.0%) |
| **Gender** | **Male (n,%)** | 53 (67.9%) |
| | **Female (n,%)** | 25(32.1%) |
| **Ethnicity** | **Arab (n,%)** | 58 (75%) |
| | **Indian (n,%)** | 15 (19%) |
| | **Asian (n,%)** | 5 (6%) |
| **Cup-to-disc ratio $\leq$ 0.3 (n,%)** | | 48 (61.5%) |
| **Diabetes Mellitus (n,%)** | | 50 (64.1%) |
| **Hyperlipidemia (n,%)** | | 40 (51.3%) |
| **Hypertension (n, %)** | | 30 (38.5%) |
| **Smoking (n,%)** | | 25 (32.1%) |
| **Ischemic heart disease (n,%)** | | 16 (20.5%) |
| **Stroke (n,%)** | | 12 (15.4%) |
| **Obstructive Sleep Apnea (n,%)** | | 9 (11.5%) |
| **Cup-to-disc ratio, mean ± SD** | | 0.32 ± 0.15 |
| **Baseline logMAR visual acuity, mean ± SD** | | 0.8 ± 0.70 |
| **Follow-up logMAR visual acuity, mean ± SD** | | 0.76 ± 0.70 |
| **Baseline visual field MD (decibels), mean ± SD** | | -15.3 ± 9.0 |
| **Follow-up visual field MD (decibels), mean ± SD** | | -13.5 ± 8.6 |
| **Follow-up duration (days), mean ± SD** | | 111 ± 44.8 |

Mean ± standard deviation.

MD = Mean Deviation.

**Table 2. Non-arteritic anterior ischemic optic neuropathy risk factor prevalence by age category (above and below 50 years of age).**

| Age Category | | Younger than 50 (N = 32,41%) | Older than 50 (N = 46,59%) | P-value |
|---|---|---|---|---|
| Gender | Male (n,%) | 27 (84.4%) | 26 (56.6%) | 0.01 * |
| | Female (n,%) | 5 (15.6%) | 20 (43.5%) | |
| Diabetes mellitus (n,%) | | 14 (43.8%) | 36 (78.3%) | 0.04 * |
| Smoking (n,%) | | 14 (43.8%) | 11 (23.9%) | 0.08 |
| Hyperlipidemia (n,%) | | 14 (35%) | 25 (65%) | 0.36 |
| Hypertension (n,%) | | 7 (21.9%) | 23 (76.7%) | 0.02 * |
| Ischemic heart disease (n,%) | | 2 (6.3%) | 14 (30.4%) | 0.01 * |
| Stroke (n,%) | | 0 | 12 (26.1%) | 0.01 * |
| Obstructive sleep apnea (n,%) | | 2 (6.3%) | 7 (15.2%) | 0.3 |
| Cup-to-disc ratio ≤ 0.3 (n,%) | | 22 (68.75%) | 26 (56.5%) | 0.3 |
| Final visual acuity equals or better than 20/40 (logMAR ≤ 0.3) (n,%) | | 17 (53.1%) | 13 (28.3%) | 0.03* |
| Improvement of ≥ 0.3 logMAR from baseline to follow up, (n,%) | | 11 (34%) | 12 (26%) | 0.4 |
| Improvement of ≥ 3 decibels in MD of visual field from baseline to follow up (n,%) | | 13 (40%) | 12 (26%) | 0.4 |

• P-value ≤0.05 is statistically significant.

females. Older NAION patients had significant higher prevalence of diabetes, hypertension, ischemic heart disease and stroke, (p <0.05). (Table 2).

However, there was a trend for higher prevalence of smoking in young NAION patients compared to older NAION patients, (p = 0.08).

## Visual outcome in young vs older NAION patients

In young patients with NAION, logMAR visual acuity improved by at least 0.3 in 11 patients (34%), remained stable in 17 patients (53.2%), and worsened in 4 patients (12.5%). Whereas in older patients with NAION, the logMAR visual acuity improved in 12 (26%), remained stable in 25 (54.3%) and worsened in 8 (19.3%). The visual field MD improved in 13 (40.6%), remained stable in 15 (46.9%) and worsened in 4 (12.5%) in the young NAION patients, whereas in older NAION patients, the MD improved in 12 (26.3%), remained stable in 28 (60.9%) and worsened 6 (13%). There was no significant difference in the mean baseline log-MAR visual acuity between young patients with NAION (0.69 +- 0.66) and older patients with NAION (0.87+- 0.72). Young NAION patients had significantly better final mean logMAR visual acuity (0.55 +- 0.57) than older NAION patients (0.9 +- 0.73), (p = 0.03). (Table 3, Fig 1).

**Table 3. Mean values of visual function at baseline and follow up based on age category (below and above age 50 years).**

| Age | Below 50 | Over 50 | P-value |
|---|---|---|---|
| Baseline logMAR Visual Acuity | 0.69 ± 0.66 | 0.87 ± 0.72 | 0.17 |
| Follow up logMAR Visual Acuity | 0.55 ± 0.57 | 0.90 ± 0.73 | 0.03 * |
| Baseline Visual Field MD | -14.13 ± 8.7 | -16 ± 9.26 | 0.35 |
| Follow up Visual Field MD | -11.52 ± 6.50 | -14.85 ± 6.70 | 0.10 |

• P-value ≤0.05 is statistically significant.

± standard deviation.

MD = Mean deviation.

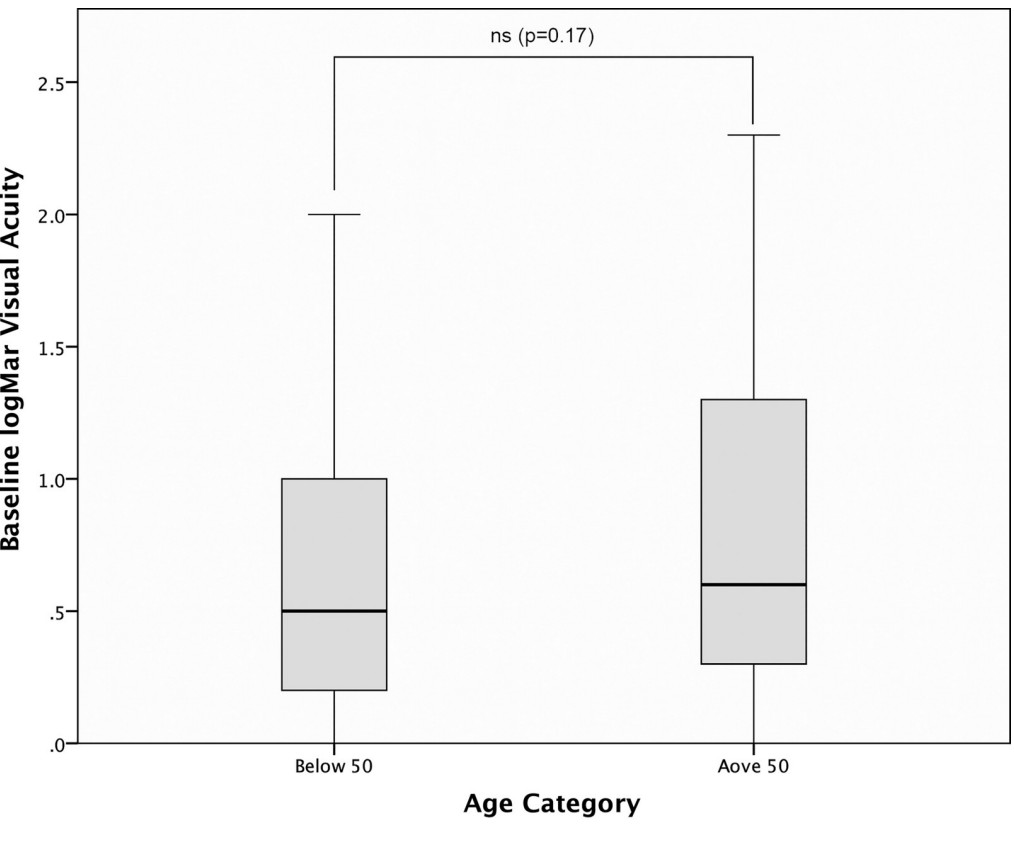

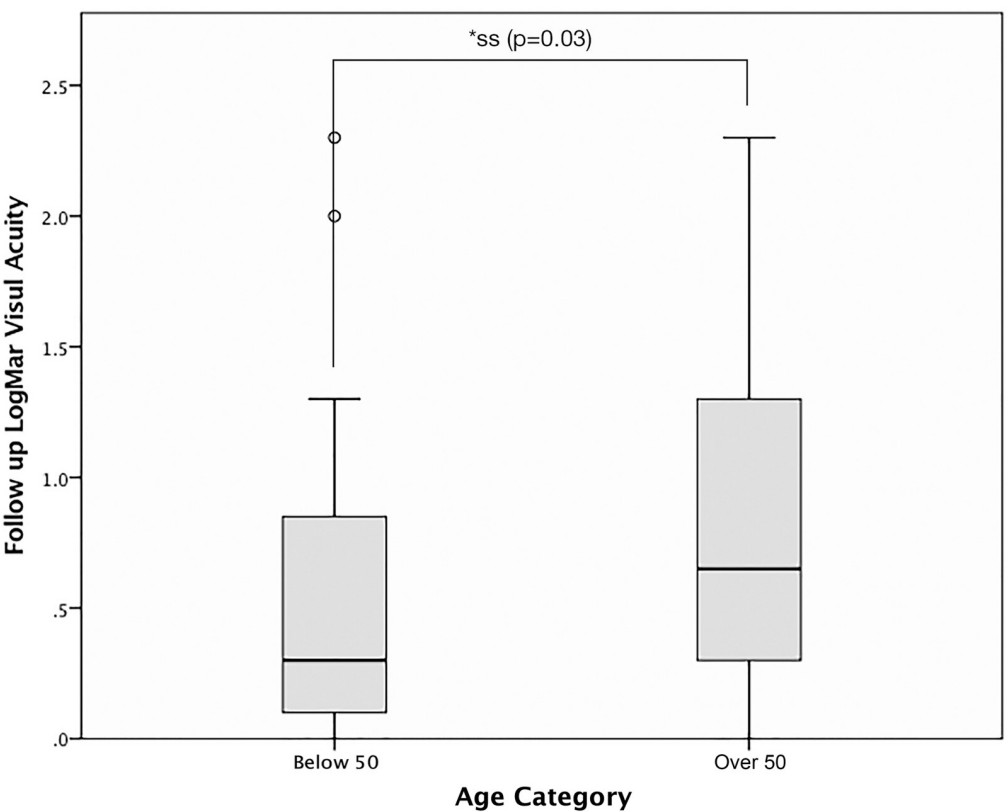

**Fig 1.** Box plot showing the comparison between baseline (A) and final (B) logMAR visual acuity between young (below 50) and older NAION (over 50). The horizontal line in the box plot represents the mean. NS = not statistically significant (p > 0.05). SS = statistically significant (p ≤0.05).

There was no significant difference in the mean MD of the visual fields between young and older NAION patients at baseline and there was a trend only towards a better mean visual field MD at follow up in young NAION patients (p = 0.1). (Table 3) Young NAION patients were 2.8 times more likely to have a final visual acuity of 20/40 or better than older NAION, odds ratio (OR), 2.87; 95% confidence interval (CI), 1.12–7.40, Chi-square p-value = 0.03).

## Effect of systemic risk factors on visual outcome

There was no significant difference in the final visual outcome between smokers vs non-smokers (p = 0.84), diabetics vs non-diabetics (p = 0.12), patients with cup-to-disc ratio ≤0.3 vs cup-to-disc ratio >0.3 (p = 0.23) in the univariate analysis. Furthermore, multivariable binary logistic regression showed that neither diabetes mellitus, smoking, ischemic heart disease, hypertension, hyperlipidemia or cup-to-disc ratio ≤0.3, had had any significant effect on attaining a final Snellen visual acuity of 20/40 or better (0.3 logMAR).

## Discussion

The most common risk factors in NAION subjects in our study was cup-to-disc ratio equal to or less than 0.3 followed by diabetes mellitus, hyperlipidemia and hypertension. (Table 1) The mean age for NAION patients in our study was 50.4 years and the subset of patients younger than 50 years comprised 41% of subjects in our study with a mean age of 41 years (range 29–48). (Table 2) This is in contrast to most large reported studies with less prevalence of NAION below the age of 50 years (10–12.5%) and the age for NAION was relatively older (around 60 years) [2,5,6,9,10]. Thirty-nine percent of young NAION patients in our study had at least one vascular risk factor associated with NAION, which reflects higher prevalence of diabetes and cardiovascular disease even among younger individuals in our region [11]. While older NAION subjects had significantly higher prevalence of diabetes, hypertension and ischemic heart disease compared to younger NAION subjects, there was a trend in the latter group towards higher prevalence of tobacco smoking. (Table 2) It has been also previously reported that smokers have an earlier age of onset of NAION, compared to former smokers and non-smokers [12].

The final mean logMAR visual acuity in young NAION patients was significantly better than older NAION patients and younger patients were significantly more likely to attain a final Snellen visual acuity of 20/40 or better. (Tables 2 and 3). There was also a trend (p = 0.1) for a better final visual field MD in younger NAION patients compared to older NAION patients. There was also a trend (p = 0.17) for better mean baseline logMAR visual acuity in young NAION patients (0.69 +- 0.66) compared to older NAION patients (0.87 +- 0.72) (Table 3). More young NAION patients than older NAION patients had at least 0.3 logMAR unit improvement in visual acuity (34% vs 26%), and 40% of young NAION patients had at least 3 decibels improvement in the visual fields MD compared to 40% of older NAION patients, but both of these changes were not statistically significant (p = 0.4). However, significantly more of young NAION patients had a final visual acuity of 20/40 or better compared to older NAION patients (53% vs 28.3%, p = 0.03) (Table 2). Preechwat et al have found also a favorable visual outcome in NAION patients younger than 50 years with 73% having a final visual acuity equal or better than 20/64 and only 13% have had a final visual acuity of less than 20/200 [8]. Sun et al., however, did not find a difference in the visual outcome in NAION

between patients younger and older than 55 years [13]. It is possible that in our subjects, young NAION patients have better final visual outcome because they tend to have a better presenting visual acuity than older NAION patients and they are also more likely to improve over the course of NAION. More studies with larger sample size are needed to investigate this further.

The most common reported systemic risk factors associated with NAION are diabetes, and hypertension [2]. Other systemic associations include hyperlipidemia, stroke, ischemic heart disease, tobacco use, systemic atherosclerosis, and obstructive sleep apnea [3,7,14]. Crowded disc (disc at risk) or small cup-to-disc ratio is another important anatomical risk factor in NAION, especially in younger patients with no systemic risk factors [15,16]. In our study, a cup-to-disc ratio ≤0.3 was seen in 61.5% overall and in 68.75% of young NAION patients and 56.5% of older NAION patients.

We could not find an independent effect of any of the systemic risk factors (diabetes, hypertension, hyperlipidemia) or optic disc configuration on the final visual outcome. Although there are several risk factors associated with NAION, the pathophysiology and the extent to which these risk factors are directly implicated in NAION is not fully understood. Current concepts about the pathophysiology of NAION is thought to involve "microvascular insufficiency" leading to edema of the optic nerve head and subsequently a compartment syndrome causing further microvascular ischemia [17]. Therefore, it is likely that it is the cumulative and inter-dependent effects of these systemic vascular risk factors which lead to optic nerve ischemia rather than any single independent effect. Sharma et al have evaluated the visual outcome in diabetic NAION versus non-diabetic patients and have shown that ischemic heart disease and greater age was associated with worse visual outcome [6]. In non-diabetic patients, the most prevalent risk factor for NAION was hyperlipidemia, while for diabetic patients, the most common risk factors were hypertension, hyperlipidemia, and small cup-to-disc ratio [6].

There are several limitations of our study including selection bias of referral to a tertiary center, the relatively small sample size, and the retrospective nature which may have led to underestimation the systemic risk factors in the study subjects. The heterogeneity of the follow up period led to variability in recording and timing the final visual outcome. However, the visual outcome (visual acuity and visual field) following NAION stabilizes quite early usually in 2–3 months and even in the so called "progressive ischemic optic neuropathy", worsening occurs in the first 2–4 weeks following NAION onset and then visual function stabilizes [18]. There is an emerging interest in the role of disc drusen in NAION especially in younger patients, and while in our study there was no patients with visible disc drusen, we did not routinely perform ocular ultrasound to rule this out. In addition, since this study spanned a long time period, newer sensitive OCT technology to detect drusen such as Enhanced-depth imaging OCT was not readily available [19].

In summary, we have found a high prevalence of systemic vascular (diabetes, hypertension, hyperlipidemia, smoking) and anatomical risk factors (cup-to-disc ratio $< = 0.3$) in patients with NAION. In our study subjects, NAION was fairly common patients under the age of 50 years and many of those patients had systemic vascular risk factors or small cup-to-disc ratio. Finally, NAION in patients below the age of 50 years tended to have a better visual outcome and were more like to have a final visual acuity of 20/40 or better than NAION patients over 50 years of age. This may be important when counseling patients with NAION about their prognosis.

## Author Contributions

**Conceptualization:** Raed Behbehani.

**Data curation:** Abdullah Ali, Ashref Al-Moosa.

**Formal analysis:** Raed Behbehani.

**Project administration:** Raed Behbehani.

**Resources:** Ashref Al-Moosa.

**Software:** Raed Behbehani.

**Supervision:** Raed Behbehani.

**Validation:** Raed Behbehani.

**Writing – original draft:** Raed Behbehani.

**Writing – review & editing:** Raed Behbehani.

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
