## [Decision Letter · Decision Letter 0]

4 Dec 2020

PONE-D-20-30757

Risk Factors and Visual Outcome of Non-Arteritic Ischemic Optic Neuropathy (NAION): Experience of A Tertiary Center in Kuwait .

PLOS ONE

Dear Dr. Behbehani,

Thank you for submitting your manuscript to PLOS ONE. After careful consideration, we feel that it has merit but does not fully meet PLOS ONE’s publication criteria as it currently stands. Therefore, we invite you to submit a revised version of the manuscript that addresses the points raised during the review process.

Both of the reviewers found the importance in your paper, so please response the problems pointed out by them.

We look forward to receiving your revised manuscript.

Kind regards,

Yoshiaki Taniyama, MD, PhD

Academic Editor

PLOS ONE

Journal Requirements:

3. Please amend your manuscript to include your abstract after the title page.

Reviewers' comments:

Reviewer's Responses to Questions

**Comments to the Author**

1. Is the manuscript technically sound, and do the data support the conclusions?

Reviewer #1: No

Reviewer #2: Yes

2. Has the statistical analysis been performed appropriately and rigorously? 

Reviewer #1: No

Reviewer #2: Yes

3. Have the authors made all data underlying the findings in their manuscript fully available?

Reviewer #1: No

Reviewer #2: Yes

4. Is the manuscript presented in an intelligible fashion and written in standard English?

Reviewer #1: Yes

Reviewer #2: Yes

5. Review Comments to the Author

Reviewer #1: The authors present characteristics of a group that has not been previously studied for NAION in the past. Thus, there is some relevance to the field.

The paper needs to be reproofed for writing, as there are typos and verbiage mistakes (‘Yong’ for young, ‘visual outcome NAION patients’, instead of ‘the visual outcome of NAION patients’), even in the abstract.

The introduction needs to be tightened and probably shortened, since they use a number of lax definitions. For example, while the ultimate lesion in NAION ultimately is an ischemic infarct, and in fact a few cases may be directly caused by ‘an infarct’, NAION is probably more commonly caused by a compartment syndrome resulting in capillary compression and ultimate ischemia, rather than a simple clot (Tesser et al). That is the reason a ‘disk at risk’ is so strongly correlated with NAION: that it predisposes to a compartment syndrome. Similarly, the studies using aspirin, IVT anti-VEGF have not shown effect, but that is not the reason treatment is directed at controlling systemic vascular risk factors. The reason treatment is directed at systemic vascular risk factors is that these have been shown to be associated with NAION. The idea that so many young individuals get NAION is quite new, and I would delve deeper into this in the introduction.

Methods:

A question comes up as to whether the younger individuals are actually experiencing NAION, and how they know this is not due to other causes such as mitochondropathy (LHON). The strongest association with diabetes is particularly interesting, since NAION is usually associated with HTN, suggesting there may be some genetic factor involved in this population. Additionally, 111 days is quite short (< 4 months), and this is a very short time to evaluate final vision in an ischemic optic nerve lesion.

Results: Table 1 shows 41% of individuals>50 years, while Table 2 shows 41% of individuals <50 years. Which is the real number?

The value of the report is based on two things: 1) the reporting from a specific regional group that has not been previously reported on. 2) the evaluation of responses and recovery in younger and older populations. The last is particularly interesting, since 59% of individuals developed NAION below 50 years of age: typically NAION is associated with individuals above the age of 50. What is the mean age of the below 50 group, and what is the youngest? This begs the question as to whether the younger individuals are actually experiencing NAION, and how they know this is not due to other causes such as mitochondropathy (LHON). They have few other associations, less crowded disks and they are male, as well as with many fewer vascular comorbidities, but strong association with smoking. The lack of visual field improvement is consistent with other studies.

The discussion needs to be rewritten, and focused on the data they have. They bring in a lot of possible associations, but little actual analysis, for the simple reason, as they point out, ‘…several limitations of our study including selection bias, etc;). The reasons for the younger individuals and their lack of correlation with classical NAION comorbidities makes me suspicious. I would really focus on a better analysis of the data that they have, evaluating the two groups better (young and old), and bring into the discussion the possibility of other (mitochondrial) disorders masquerading as NAION, particularly in the younger individuals, as well as performing a secondary analysis on these younger individuals, with possible mitochondrial screening if available (it may not be). The reason for this caution is that the authors are potentially bringing into the literature a report that people will quote about ‘NAION affects more young people than people over 50’, and this is going to be very confusing, and possibly wrong. In this case, I do not think it excessive caution.

Reviewer #2: Comments:

This is a retrospective study which determined systemic vascular risk factors and disc configurations in NAION patients. The authors also compared visual function outcome between young and older subgroups.

1.“a crowded disc or small cup-to-disc ratio (=<0.3) was found in 31% of young and 43.3% of older NAION.” Do you include either crowded disc or small cup-to-disc ratio? I didn’t find the definition of crowded disc, small C/D, or disc at risk in methods? Is “Small C/D ratio” equal to or smaller than 0.3? This should be defined in methods about how you recruited the patients. But 0.3 is usually considered as normal ratio. “Disc at risk” is usually appreciated in the unaffected eye, because the affected optic nerve head is swollen. A “Disc at Risk” commonly has a cup to Disc Ratio less than 0.3 (usually 0.1). “Crowded disc” is usually used to describe small optic nerve head. The definition you use for the analysis should be clarified. It would be better to cite references.

2. Introduction Line 4: “Although NAION is thought to be infarction of the optic nerve head”. Based on multiple papers, NAION is thought to be associated with “hypoperfusion of optic nerve head” rather than infarction.

3. “Only one eye of each new NAION case was included” Is this for bilateral NAION? The number of unilateral AION and bilateral AION should be clarified.

4. “patients with severe diabetic retinopathy” Please define the “severe DR”. Is it defined by international clinical classification system for DR and DME? Does this mean the study excluded the DR that is worse than or equal to “severe NPDR”?

5. Visual field MD should be spelled out when first appeared in the text.

6. “Visual acuity change was defined as 0.1 logMAR unit change was considered either improvement or worsening.” Any references to support this definition? There is a paper recommended using at least 0.2 logMAR or greater to distinguish acuity changes from no change. (Daniel A. Rosser. How Sensitive to Clinical Change are ETDRS logMAR Visual Acuity Measurements? IOVS. 2003; 44(8): 3278-3281.) In Dr. Hayreh’s paper (reference 9), they used 0.3 logMAR.

7. Results: Seventy right eyes should be “ Seventy-eight”?

8. Table 1 should have headers for each column on the top. (e.g. Variables)

9. I recommend ranking the variables in table 1 and 2 in reasonable order, e.g. from the most common to the least common risk factors (diabetes, hyperlipidemia, hypertension, smoking…). You can still separate the systemic factors from ocular configurations and visual functions as it is now.

10. I’m curious about how many patients had finished 3 months and 6 months follow-up. Two months follow up is relatively short term.

11. Recently some papers discussed about young NAION and optic disc drusen. Did you have any chance to rule out optic disc drusen in young NAION in this study?

12. I don’t really understand this sentence. “except four patients who had 1 month of follow up and excluding those patients those patients had no effect on the final visual outcome analysis.” Who (how many) are included and who (how many) are excluded? If you didn’t include all subjects for follow-up analysis, I recommend mentioning it in table 1 as a footnote. I assume all the data in table 1 should be collected from 78 patients (eyes).

13. I recommend separating the results into several paragraphs with headlines. E.g. Basic characteristics of NAION patients (This should include follow-up), The risk factors in NAION patients, The risk factors in young NAION vs. older NAION, The visual function outcome in young NAION vs. older NAION…

14. I also recommend having diabetic or nondiabetic patients as subgroups to further analysis (risk factors and visual function).

15. “There was no statistically significant difference in the MD of the visual fields between young and older NAION patients neither at baseline nor follow up. ” Do you mean “no significant difference …either at baseline or follow-up”?

16. In the first and the second paragraphs of discussion, the “ischemic optic decompression trial” is the same as “In the optic nerve sheath decompression trial (IONDT) ”. This should be consistent with “IONDT”.

17. The heterogeneity of the follow up period meant the method of recording the final visual outcome was not standardized. In my opinion, the “methods of recording visual outcome” are visual field and acuity test but not follow up period.

18. In figure 1(A), ns should be followed by p value even if p > 0.05.

19. For table 2, I recommend calculating the p value for comparison of gender (male vs. female).

6. PLOS authors have the option to publish the peer review history of their article (what does this mean?). If published, this will include your full peer review and any attached files.

Reviewer #1: No

Reviewer #2: No

---

## [Author Response · Author response to Decision Letter 0]

14 Dec 2020

Answers to Reviewer's Comments 

Reviewer #1: The authors present characteristics of a group that has not been previously studied for NAION in the past. Thus, there is some relevance to the field.

The paper needs to be reproofed for writing, as there are typos and verbiage mistakes ('Yong' for young, 'visual outcome NAION patients', instead of 'the visual outcome of NAION patients'), even in the abstract.

The introduction needs to be tightened and probably shortened, since they use a number of lax definitions. For example, while the ultimate lesion in NAION ultimately is an ischemic infarct, and in fact a few cases may be directly caused by 'an infarct', NAION is probably more commonly caused by a compartment syndrome resulting in capillary compression and ultimate ischemia, rather than a simple clot (Tesser et al). That is the reason a 'disk at risk' is so strongly correlated with NAION: that it predisposes to a compartment syndrome. Similarly, the studies using aspirin, IVT anti-VEGF have not shown effect, but that is not the reason treatment is directed at controlling systemic vascular risk factors. The reason treatment is directed at systemic vascular risk factors is that these have been shown to be associated with NAION. The idea that so many young individuals get NAION is quite new, and I would delve deeper into this in the introduction.

Answer: The introduction was shortened and some sentences were removed, notably the sentence stating that the etiology is "infarction of the optic nerve". We did not mean to imply that the etiology was due to an embolus and we have elaborated a bit, as suggested by the reviewer, that the current understanding of NAION is "hypo-perfusion" of the optic nerve head with secondary swelling and a "compartment syndrome". We agree that that some of the treatments stated (IVA, steroids) are of limited success and that sentence was removed. Furthermore, we have added the reference suggested by the reviewer (Tesser et al. The morphology of infarct in NAION) in the introduction. 

Methods:

A question comes up as to whether the younger individuals are actually experiencing NAION, and how they know this is not due to other causes such as mitochondropathy (LHON). The strongest association with diabetes is particularly interesting, since NAION is usually associated with HTN, suggesting there may be some genetic factor involved in this population. Additionally, 111 days is quite short (< 4 months), and this is a very short time to evaluate final vision in an ischemic optic nerve lesion.

Answer: It is widely-accepted that NAION is a clinical diagnosis made based on the clinical history and examination findings and is associated with a characteristic clinical features natural history. While other optic nerve disorders such LHON can mimic NAION, it is virtually impossible to rule out LHON in every patient unless the genetic testing in every single case. The high prevalence of diabetes of NAION can be explained by the high prevalence of diabetes in the general population in Kuwait, even among younger patients. Finally, we think that the follow up period (111 days) is relatively short but still adequate to assess the final visual outcome in NAION patients since vision (visual acuity, visual field) does not change significantly following 2-3 months, but the disc changes (optic disc swelling and subsequent atrophy) does lag behind and may take longer to finally stabilize. (N R Miller, A C Arnold. Current concepts in the diagnosis, pathogenesis and management of Non-arteritic anterior ischemic optic neuropathy. Eye. 2015) 

Results: Table 1 shows 41% of individuals>50 years, while Table 2 shows 41% of individuals <50 years. Which is the real number?

Answer: Patients younger than 50 composed 41% and this was corrected in table 1. 

The value of the report is based on two things: 1) the reporting from a specific regional group that has not been previously reported on. 2) the evaluation of responses and recovery in younger and older populations. The last is particularly interesting, since 59% of individuals developed NAION below 50 years of age: typically NAION is associated with individuals above the age of 50. What is the mean age of the below 50 group, and what is the youngest? This begs the question as to whether the younger individuals are actually experiencing NAION, and how they know this is not due to other causes such as mitochondropathy (LHON). They have few other associations, less crowded disks and they are male, as well as with many fewer vascular comorbidities, but strong association with smoking. The lack of visual field improvement is consistent with other studies.

Answer: We thank the reviewer for this comment. Patients younger than 50 years comprised 41% of all NAION patients in the study as stated above. The mean age of patients below 50 years was 41 years and the minimum age was 29 and the maximum was 48. We agree that the is a strong trend in our study in young NAION patients with Tobacco consumption, and although it did not reach statistical significance this may have been due to relatively low sample size. The association between tobacco use and an earlier age of onset of NAION has already been reported (Hayreh et al. Nonarteritic anterior ischemic optic neuropathy and tobacco smoking. Ophthalmology , 2007) and we have added this reference in the manuscript. 

The discussion needs to be rewritten, and focused on the data they have. They bring in a lot of possible associations, but little actual analysis, for the simple reason, as they point out, '…several limitations of our study including selection bias, etc;). The reasons for the younger individuals and their lack of correlation with classical NAION comorbidities makes me suspicious. I would really focus on a better analysis of the data that they have, evaluating the two groups better (young and old), and bring into the discussion the possibility of other (mitochondrial) disorders masquerading as NAION, particularly in the younger individuals, as well as performing a secondary analysis on these younger individuals, with possible mitochondrial screening if available (it may not be). The reason for this caution is that the authors are potentially bringing into the literature a report that people will quote about 'NAION affects more young people than people over 50', and this is going to be very confusing, and possibly wrong. In this case, I do not think it excessive caution.

Answer: As suggested by the reviewer, the discussion was shortened to focus on the data analysis and we have removed parts of the discussion which may not be relevant to our study. We have discussed the possible association between NAION and tobacco use in younger patients. Unfortunately, since this is a retrospective study, mitochondrial screening for younger NAION is not feasible and would be out of the scope of the study. Although LHON although occurs more frequently in young patients, it can present at any age even in older patients and the presence and thus this would require another study possibly perhaps looking at the association between NAION and LHON mutations. Finally, many of the mitochondrial mutations if detected can be due to genetic heterogeneity and polymorphism and not necessarily causing LHON. We believe that NAION is a clinical diagnosis and is being diagnosed more in younger patients. The strong trend for tobacco use in our younger patients with NAION in our study may also explain the relatively high prevalence of NAION. 

Reviewer #2: Comments:

This is a retrospective study which determined systemic vascular risk factors and disc configurations in NAION patients. The authors also compared visual function outcome between young and older subgroups.

1."a crowded disc or small cup-to-disc ratio (=<0.3) was found in 31% of young and 43.3% of older NAION." Do you include either crowded disc or small cup-to-disc ratio? I didn't find the definition of crowded disc, small C/D, or disc at risk in methods? Is "Small C/D ratio" equal to or smaller than 0.3? This should be defined in methods about how you recruited the patients. But 0.3 is usually considered as normal ratio. "Disc at risk" is usually appreciated in the unaffected eye, because the affected optic nerve head is swollen. A "Disc at Risk" commonly has a cup to Disc Ratio less than 0.3 (usually 0.1). "Crowded disc" is usually used to describe small optic nerve head. The definition you use for the analysis should be clarified. It would be better to cite references.

Answer: To avoid confusion, we have removed the terms "crowded disc" or "disc at risk" and we have replaced that with "cup-to-disc ratio <=0.3" to be all encompassing since as the reviewer has pointed out that a disc-at-risk or crowded disc have much smaller cup-to-disc ratio and the sometimes the cup is totally absent. Some textbooks would cite 0.4 to 0.5 as "normal" and below that would be "low" although and in the AAO manual for glaucoma it is stated that "C/D ratio between 0.1-0.4 can be normal" while most clinicians would perhaps 0.1 as low. Finally, we have obtained vertical C/D ratio by OCT and it is understandable that there may be discordance as C/D ratio can be under-estimated with ophthalmoscopy alone. 

2. Introduction Line 4: "Although NAION is thought to be infarction of the optic nerve head". Based on multiple papers, NAION is thought to be associated with "hypoperfusion of optic nerve head" rather than infarction.

Answer: This has been addressed and the introduction was changed. 

3. "Only one eye of each new NAION case was included" Is this for bilateral NAION? The number of unilateral AION and bilateral AION should be clarified.

Answer: This sentence was removed to avoid confusion. We have included only unilateral NAION and patient with prior NAION in the contralateral eye were excluded. 

4. "patients with severe diabetic retinopathy" Please define the "severe DR". Is it defined by international clinical classification system for DR and DME? Does this mean the study excluded the DR that is worse than or equal to "severe NPDR"?

Answer: Yes. We have excluded patents severe non-proliferative diabetic retinopathy/proliferative DR and macular edema. This was corrected and clarifies in the text.

5. Visual field MD should be spelled out when first appeared in the text.

Answer: This was corrected. 

6. "Visual acuity change was defined as 0.1 logMAR unit change was considered either improvement or worsening." Any references to support this definition? There is a paper recommended using at least 0.2 logMAR or greater to distinguish acuity changes from no change. (Daniel A. Rosser. How Sensitive to Clinical Change are ETDRS logMAR Visual Acuity Measurements? IOVS. 2003; 44(8): 3278-3281.) In Dr. Hayreh's paper (reference 9), they used 0.3 logMAR.

Answer: We have corrected this typo error and indeed we have used 0.3 logmar change as the minimum for change for either improvement or worsening as in the study by Hayreh et al. Nonarteritic Anterior Ischemic Optic Neuropathy: Natural History of Visual Outcome. Ophthalmology 2008. 

7. Results: Seventy right eyes should be " Seventy-eight"?

Answer: corrected.

8. Table 1 should have headers for each column on the top. (e.g. Variables)

Answer: This was added. 

9. I recommend ranking the variables in table 1 and 2 in reasonable order, e.g. from the most common to the least common risk factors (diabetes, hyperlipidemia, hypertension, smoking…). You can still separate the systemic factors from ocular configurations and visual functions as it is now.

Answer: This was done.

10. I'm curious about how many patients had finished 3 months and 6 months follow-up. Two months follow up is relatively short term.

Answer: Fifty out of the 75 patients (67%) in of our study had at least 3-month follow up while the rest (33%) had at least had at least 2-month follow up. Only four patients had 1month follow up and when excluded from the in the data analysis as a trial there was no effect on the results so we have included then in the final analysis. We believe that 2-3 months follow up period is adequate for NAION since the final visual outcome tends to stabilize relatively early in NAION (2-3 months) as opposed to other optic neuropathies (N R Miller, A C Arnold. Current concepts in the diagnosis, pathogenesis and management of nonarteritic anterior ischaemic optic neuropathy. Eye. 2015), but in the discussion we did acknowledge this as one of the study limitations. 

11. Recently some papers discussed about young NAION and optic disc drusen. Did you have any chance to rule out optic disc drusen in young NAION in this study?

Answer: That would be an interesting point and although none of the patients in our study had visible drusen, we did not look for buried drusen by Ultrasound. Furthermore, the study spanned a long period of time, at which perhaps new technology such as EDI-OCT, which are very sensitive and reliable to rule out disc drusen was not available. We have mentioned this also as one of study limitations in the discussion. 

12. I don't really understand this sentence. "except four patients who had 1 month of follow up and excluding those patients those patients had no effect on the final visual outcome analysis." Who (how many) are included and who (how many) are excluded? If you didn't include all subjects for follow-up analysis, I recommend mentioning it in table 1 as a footnote. I assume all the data in table 1 should be collected from 78 patients (eyes).

Answer: This sentence was removed to avoid confusion. We have included all patients in the table and follow up analysis. When we excluded the 4 patients from as trial in the data analysis, there was no change in the statistical significance of the results so we decided to include them since there was no benefit of excluding them. 

13. I recommend separating the results into several paragraphs with headlines. E.g. Basic characteristics of NAION patients (This should include follow-up), The risk factors in NAION patients, The risk factors in young NAION vs. older NAION, The visual function outcome in young NAION vs. older NAION…

Answer: This was done as suggested by the reviewer. 

14. I also recommend having diabetic or nondiabetic patients as subgroups to further analysis (risk factors and visual function).

Answer: We have performed both uni-variable and multivariable logistic regression analysis to determine if ant of the prevalent systemic risk factors (smoking, diabetes, hyperlipidemia) or ocular (small cup-to-disc ratio) had any effect on the final visual outcome and we could not see any. This was mentioned in the last paragraph in the results section but we have elaborated on it more in the revision.

15. "There was no statistically significant difference in the MD of the visual fields between young and older NAION patients neither at baseline nor follow up. " Do you mean "no significant difference …either at baseline or follow-up"?

Answer: Yes. The difference did not reach statistical significance. The sentence was reworded as suggested to avoid confusion.

16. In the first and the second paragraphs of discussion, the "ischemic optic decompression trial" is the same as "In the optic nerve sheath decompression trial (IONDT) ". This should be consistent with "IONDT".

Answer: We have corrected this as suggested and this study was abbreviated as IONDT later throughout the text. 

17. The heterogeneity of the follow up period meant the method of recording the final visual outcome was not standardized. In my opinion, the "methods of recording visual outcome" are visual field and acuity test but not follow up period.

Answer: This was changed to "timing of recording the final visual outcome was not standardized".

18. In figure 1(A), ns should be followed by p value even if p > 0.05.

Answer: We have added this as suggested.

19. For table 2, I recommend calculating the p value for comparison of gender (male vs. female).

Answer: This was done and added as suggested.

---

## [Decision Letter · Decision Letter 1]

19 Jan 2021

PONE-D-20-30757R1

Risk Factors and Visual Outcome of Non-Arteritic Ischemic Optic Neuropathy (NAION): Experience of A Tertiary Center in Kuwait .

PLOS ONE

Dear Dr. Behbehani,

Thank you for submitting your manuscript to PLOS ONE. After careful consideration, we feel that it has merit but does not fully meet PLOS ONE’s publication criteria as it currently stands. Therefore, we invite you to submit a revised version of the manuscript that addresses the points raised during the review process.

One of the reviewer wants the authors to correct the manuscript by a native speaker.

We look forward to receiving your revised manuscript.

Kind regards,

Yoshiaki Taniyama, MD, PhD

Academic Editor

PLOS ONE

Reviewers' comments:

Reviewer's Responses to Questions

**Comments to the Author**

1. If the authors have adequately addressed your comments raised in a previous round of review and you feel that this manuscript is now acceptable for publication, you may indicate that here to bypass the “Comments to the Author” section, enter your conflict of interest statement in the “Confidential to Editor” section, and submit your "Accept" recommendation.

Reviewer #1: (No Response)

Reviewer #2: All comments have been addressed

2. Is the manuscript technically sound, and do the data support the conclusions?

Reviewer #1: No

Reviewer #2: Yes

3. Has the statistical analysis been performed appropriately and rigorously? 

Reviewer #1: No

Reviewer #2: Yes

4. Have the authors made all data underlying the findings in their manuscript fully available?

Reviewer #1: Yes

Reviewer #2: Yes

5. Is the manuscript presented in an intelligible fashion and written in standard English?

Reviewer #1: No

Reviewer #2: Yes

6. Review Comments to the Author

Reviewer #1: The paper has bad spelling, grammar, to the point that it is painful to read and it was not reviewed by a native English speaker. The authors must understand that this is being read by a non-specialist audience who will not stop to struggle through their syntax, in addition to the errors they made in their arguments. I STRONGLY recommend that the manuscript should be reviewed by someone whose primary language is English before this is resubmitted! There are some important data here but it is overshadowed by non-scientific issues! There are also problems with both understanding of facts and logic! Please see my other comments in the attached review

Reviewer #2: Thanks for the rapid response and the authors had addressed all the questions.

Only some minor comments:

1. Table 1: The heading on the right column should be" Patients (n=78)". Second row, Age (years, mean± SD). Third row, "Below 50 years (n, %)". Please correct all the others in this table.

2. "+_"should be"±". "<=" should be "≤".

3. Results part 2, revise the subheading: Visual Outcome in Young NAION Versus Older NAION

7. PLOS authors have the option to publish the peer review history of their article (what does this mean?). If published, this will include your full peer review and any attached files.

Reviewer #1: No

Reviewer #2: No

---

## [Author Response · Author response to Decision Letter 1]

21 Jan 2021

Answers to reviewer Comments

Reviewer 1:

1) The paper has bad spelling, grammar, to the point that it is painful to read. I STRONGLY recommend that the manuscript should be reviewed by someone whose primary language is English before this is resubmitted!

Answer: The paper was revised extensively again for syntax, grammar and spelling errors and the errors pointed out by the reviewer and other errors found in the text were corrected.

2) Glaucoma is the most common optic neuropathy (it affects 60,000,000 people in the world, vs ~10,000,000 max for NAION, assuming an incidence of 10.2/100,000 for NAION). NAION is the second most common optic neuropathy and the most common cause for sudden optic nerve-related vision loss.

Answer: Agree. We have added the sentence suggested by the reviewer.

3) Put the reference numbers (1); ((2,3) before the period at the end of the sentence.

Answer: This was changed throughout the text.

4) Results: The paragraph describing visual outcome in young vs old NAION is confusing and should be revised (and this should be written: young patients (or individuals) vs old patients (or individuals) with NAION). The last sentence in this paragraph is describing the MEAN final logmar visual acuity for Young individuals with NAION and MEAN final logmar visual acuity for old individuals with NAION.

Answer: This was changed and corrected as suggested by the reviewer.

5) The authors need to clarify their discussion of the effect of improved mean acuity in the young individuals with NAION (paragraph 2). The reason for this is the difference between the number of younger individuals who improve their acuity, vs the MEAN final acuity. There is already a trend for better acuity in young individuals with NAION vs older individuals with NAION (Baseline mean logMAR visual acuity; p value=0.17). Young NAION patients: Old NAION patients:

 Visual acuity improved in 21.9% (n=7), Visual acuity improved in 39% (n=18)

 Stable in 56.3% (n=18) Stable in 33.6% (n=15)

 Worsened in 21.9% (n=7) Worsened in 28.3% (n=13)

That the ULTIMATE MEAN visual acuity in the younger age group is better does not mean that the outcome for any younger individual is more favorable, since fewer younger individuals have improvements in their visual acuity. The authors are clearly trying to puzzle this out with their second paragraph, with their inclusion of Sun, Hayreh and IONDT data.

Answer: We thank the reviewer for this comment. I think we were responsible for this confusion since the we have the percentages above was part of overview descriptive statistics and ANY change in visual acuity even single Snellen line or 0.1 logMAR unit was calculated as improvement in visual acuity. This has to led to over-estimation of improvement or worsening in visual acuity and was reflected by having seemingly more improvement in visual acuity in older NAION patients. However, when we re-calculated this based on the criteria of for visual acuity improvement presented in the methods section (improvement of visual acuity of at least 0.3 logMAR unit), we have found that slightly more of young NAION patients actually had improved visual acuity and the majority of patients in both young and older NAION patients was stable. The revised calculation is the following:

LogMAR acuity change Young NAION patients Older NAION patients

Better 11 (34%) 12 (26%) 

Stable 17 (53.2%) 25 (54.3%)

Worse 4 (12.5%) 8 (19.3%)

Similarly, when have calculated a change of MD visual field as at least 3 decibel change to qualify for either improvement or worsening in MD, we found the following:

MD visual field change Young NAION patients Older NAION patients

Better 13 (40.6%) 12 (26.3%)

Stable 15 (46.9%) 28 (60.0%)

Worse 4 (12.5%) 6 (13%)

Therefore, despite having a trend for younger NAION patients for a better presenting visual acuity compared to older NAION patients, more of the young NAION patients had an improvement of logMAR visual acuity than older NAION patients from baseline to follow up (34% vs 26%) but this was not statistically significant. Similarly, more of the young NAION patients had improvement of the visual field MD than older NAION patients from baseline to follow up (40.6% vs 26.3%) but this was also not statistically significant. Only when we used a final visual acuity of 20/40 or better as an outcome, young NAION patients reached this outcome significantly more than older NAION patients. Finally, there was a trend towards young NAION patients having a better visual field MD at follow up than older NAION patients (-11.52 vs 14.85, p=0.1). 

Therefore, in trying to answer the question posed by the reviewer as to whether young NAION patients have a better final visual acuity because there was a trend towards in them having a better presenting visual acuity, this certainly may one valid hypothesis. However, at least by numbers, young NAION patients also they also were more likely to improve in visual acuity (34% vs 26%) and visual field MD (40.6% vs 26.3%) from baseline to follow up. Therefore, it is quite possible that they there is an element both factors (better presenting visual acuity and higher likelihood of improvement over the course of NAION in younger patients) but our study was not powered enough for this to reach statistical significance. We have added this clarification in the discussion and we have acknowledged this as a limitation of our study. We have updated table 2 to include these comparisons. Finally, we have removed the sentences in the discussion citing the paper Hayreh and the IODNT which we felt they were strictly non-relevant to our findings.

6) Importantly, the authors have made a crucial error in their third paragraph. A cup to disk ratio <0.3 cannot be seen in 61.5% of patients overall when you have 31% of young individual with a c:d <0.3 and 43% of older individuals with a c:d<0.3. The total number of individuals was 78; the total number of individuals with C:D< 0.3 is 30; the percentage of total with C:D<0.3 is 38.4% (30/78).

Answer: We thank the reviewer for this comment. This has unfortunately stemmed from an inadvertent error in Table 2 which we have corrected since the total number of patients in the with c:d ratio <0.3 is 48 as presented in table 1 and numbers table 2 before correction was 10 in young NAION patients and 20 in older NAION patients (total 30). The true number of young NAION patients with c:d <0.3 is 22 of total 32 patients (68.75%) and in older NAION patients it is 26 of total 46 patients (56.5%) . So the total number of patients with c:d <0.3 is (22+26=48) and is 61.5% (48/78) of the total subjects.

Reviewer 2

1. Table 1: The heading on the right column should be" Patients (n=78)". Second row, Age (years, mean± SD). Third row, "Below 50 years (n, %)". Please correct all the others in this table.

Answer: All table headings were revised and corrected as suggested.

2. "+_"should be"±". "<=" should be "≤".

Answer: This was changed as suggested throughout the text.

3. Results part 2, revise the subheading: Visual Outcome in Young NAION Versus Older NAION

Answer: This was changed to "Visual Outcome in Young vs Older NAION Patients".

---

## [Decision Letter · Decision Letter 2]

2 Feb 2021

Risk Factors and Visual Outcome of Non-Arteritic Ischemic Optic Neuropathy (NAION): Experience of A Tertiary Center in Kuwait .

PONE-D-20-30757R2

Dear Dr. Behbehani,

We’re pleased to inform you that your manuscript has been judged scientifically suitable for publication and will be formally accepted for publication once it meets all outstanding technical requirements.

Kind regards,

Yoshiaki Taniyama, MD, PhD

Academic Editor

PLOS ONE

Additional Editor Comments (optional):

Reviewers' comments:

Reviewer's Responses to Questions

**Comments to the Author**

1. If the authors have adequately addressed your comments raised in a previous round of review and you feel that this manuscript is now acceptable for publication, you may indicate that here to bypass the “Comments to the Author” section, enter your conflict of interest statement in the “Confidential to Editor” section, and submit your "Accept" recommendation.

Reviewer #1: All comments have been addressed

Reviewer #2: All comments have been addressed

2. Is the manuscript technically sound, and do the data support the conclusions?

Reviewer #1: Yes

Reviewer #2: Yes

3. Has the statistical analysis been performed appropriately and rigorously? 

Reviewer #1: Yes

Reviewer #2: Yes

4. Have the authors made all data underlying the findings in their manuscript fully available?

Reviewer #1: Yes

Reviewer #2: Yes

5. Is the manuscript presented in an intelligible fashion and written in standard English?

Reviewer #1: Yes

Reviewer #2: No

6. Review Comments to the Author

Reviewer #1: Page 19 of the compiled PDF: add the words to 'therefore it is likely that IT IS the cumulative and inter-dependent effects.....

Reviewer #2: Thanks for the authors addressing all the comments. There are still some errors that need to be revised. I hope the whole paper can be polished before accepted.

1. “Non-arteritic ischemic optic neuropathy (NAION) is the second most common optic neuropathy following glaucoma in patients over the age of 50 years most common cause for sudden optic nerve-related vision loss.” This is not a sentence. Please revise it. “Optic neuropathy presenting with sudden vision loss” or “ optic nerve-related acute vision loss” is better than “sudden optic nerve-related vision loss”.

2. In results, “predominantly Middle-Eastern Arabs followed by Indian and Asian backgrounds.” I would recommend giving the exact numbers if it is available.

3. There are still some grammatical errors and the errors of putting space before or after the punctuation. e.g. It is associated with systemic risk factors such as diabetes , hypertension , hyperlipemia and anatomical risk factors mainly crowded anomalous optic disc (disc at risk) or small cup-to-disc ratio (2, 3). Please delete the space after diabetes.

4. Please check “+-” throughout the whole paper.

5. In table 1, “Cup-to-disc ratio , mean± SD”，Space should be added before “±”.

6. “<=” should be “≤”.

7. PLOS authors have the option to publish the peer review history of their article (what does this mean?). If published, this will include your full peer review and any attached files.

Reviewer #1: No

Reviewer #2: No

---

## [Editor Report · Acceptance letter]

3 Feb 2021

PONE-D-20-30757R2 

Risk Factors and Visual Outcome of Non-Arteritic Ischemic Optic Neuropathy (NAION): Experience of A Tertiary Center in Kuwait.  

Dear Dr. Behbehani:

I'm pleased to inform you that your manuscript has been deemed suitable for publication in PLOS ONE. Congratulations! Your manuscript is now with our production department. 

Kind regards, 

on behalf of

Dr Yoshiaki Taniyama 

Academic Editor

PLOS ONE